# Clinical and Epidemiological Characteristics of Patients with COVID-19 Admitted to the Intensive Care Unit: A Two-Year Retrospective Analysis

**DOI:** 10.3390/life13030741

**Published:** 2023-03-09

**Authors:** Juliana Raimondo e Silva Malzone, Ana Paula Ribeiro, Tatiane Silva de Souza, Debora Driemeyer Wilbert, Neil Ferreira Novo, Yara Juliano

**Affiliations:** 1Health Science Post-Graduate Department, School of Medicine, University Santo Amaro, São Paulo 04829-300, Brazil; 2Physical Therapy Department, School of Medicine, University of Sao Paulo, São Paulo 05360-160, Brazil

**Keywords:** COVID-19, coronavirus, ventilation, mortality

## Abstract

In March 2020, COVID-19 was characterized as a pandemic by the World Health Organization. Hospitalized patients affected by COVID-19 presented with severe respiratory and motor impairment, especially those who required intensive treatment and invasive mechanical ventilation, with sequelae that extended after the period of hospitalization. Thus, the aim of the current study was to verify the clinical and epidemiological characteristics of patients with COVID-19 admitted to the Intensive Care Unit in 2020 and 2021, according to age group. Methods: A retrospective cohort study. Data were collected through the “ICUs Brasileiras” between March 2020 and November 2021 for severe acute respiratory syndrome (SARS) due to COVID-19. The following were analyzed: the number of hospital admissions, days in the ICU and hospital, clinical aspects (non-invasive or invasive ventilatory support, comorbidities, frailty, SAPS 3 and SOFA severity scales, use of amines and renal support), and ICU and hospital mortality rate. Results: A total of 166,966 ICU hospital admissions were evaluated over the evaluated quarters. The main results showed a peak in the number of hospitalizations between March and May 2021, with a higher percentage of males. The peak of ICU admissions for 7 days was between March and May 2021 and 21 days between March and May 2020. In addition, higher deaths were observed in the age groups between 40 and 80 years between 2020 and 2021, with the group above 81 being the age group with the highest mortality. Mortality in the ICU of ventilated patients was higher in the age group above 70 years. Another observation was the predominance of SAPS 3 and the peak of mechanical ventilation for more than 7 days between June and August 2021. Conclusion: The clinical and epidemiological characteristics of patients with COVID-19 were influenced by age group, showing higher mortality over 81 years and over 70 years in the ICU supported by mechanical ventilation, maintained for 7 days from June to August 2021. The years 2020 and 2021 also showed differences for patients with COVID-19, with greater hospitalization between March and May 2021, especially in the ICU for 7 days, and between March and May 2020 for the 21-day period.

## 1. Introduction

The new coronavirus pandemic that we are currently experiencing began in the city of Wuhan, China, on December 31, 2019, when an outbreak of pneumonia cases was reported for the first time to the World Health Organization (WHO) [1,2]. In 2020, the virus was named the severe acute respiratory syndrome coronavirus—2 (SARS-CoV-2), the etiologic agent causing coronavirus disease 2019 (COVID-19). In March 2020, COVID-19 was characterized as a pandemic by the WHO [2]. SARS-CoV-2 belongs to the same subgenus as severe acute respiratory Syndrome (SARS), which caused an epidemic in China in 2003 [3]. The disease presents a high power of transmissibility, mainly through respiratory droplets, aerosols, and through direct contact [4].

The disease has a varied presentation, from an absence of symptoms to generalized interstitial pneumonia associated with acute respiratory distress syndrome (ARDS), resulting in a mortality rate of 5 to 10% [4]. In Brazil, on 26 February 2020, the first patient, returning to São Paulo from a trip to northern Italy, received a confirmed diagnosis of COVID-19 in São Paulo [5]. Approximately 20% of individuals with COVID-19 develop the severe form of the disease, requiring hospitalization and oxygen therapy support, associated with lung injury and hypoxemic respiratory failure. Around 5% are admitted to the Intensive Care Unit (ICU), where they need invasive mechanical ventilation for the treatment of ARDS, followed by an invasion of the virus into the central nervous system, including the brainstem, followed by acute respiratory distress syndrome as a later complication [6]. For this reason, a personalized respiratory and monitoring strategy in patients with SARS-CoV-2 infection is paramount. An individualized ventilatory approach based on lung physiology, morphology, imaging, and identification of biological phenotypes may improve COVID-19 outcomes while individualizing mechanical ventilation practices [6,7].

The consequences caused by COVID-19 are diverse, affecting several systems, with the respiratory system being one of the most commonly affected, regardless of the symptomatology. Respiratory sequelae are varied and depend on the type of injury, associated infections, and presence of previous pulmonary comorbidities, highlighting a reduction in lung capacity and volume, changes in radiographic findings, limitations in the performance of exercises, and, consequently, a decrease in functional capacity. According to Skopljanac et al., in 2022 [8], a predictive model identified four key parameters that can predict an adverse COVID outcome during the patient admission to the hospital; LUS score, day of illness, leukocyte count, and presence of cardiovascular disease. In addition, a reticular pattern with a posterior distribution was observed three months after discharge from severe COVID-19 pneumonia, and this differs from previously described postCOVID-19 fibrotic-like changes [9].

According to the literature, therapeutic assistance through physiotherapy is paramount to avoid sequelae and diseases arising from COVID-19, made up of organized and systematized assistance to restore the functionality of patients affected by severe illness. Therefore, the physiotherapist should outline the therapeutic plan individually, with different techniques applied based on the individuality and needs of each patient [8]. In addition, it is widely known that in around 30 to 60% of cases, ICU admissions are associated with prolonged immobilization, with consequent loss of muscle mass and, in turn, a decrease in strength and functionality [10,11].

In clinical and physiotherapeutic treatment, oxygen therapy is a priority, being one of the clinical pillars; however, patients who remain hypoxemic or continue to present signs of respiratory distress may require the use of a mask with a non-rebreathing oxygen reservoir, non-invasive ventilation (NIV), or a high-flow nasal catheter (HFNC). In patients unable to maintain adequate ventilation levels using non-invasive measures, intubation and invasive mechanical ventilation (IMV) are required to ensure adequate tissue oxygenation [8]. SARS is characterized by acute diffuse lung injury that leads to increased pulmonary vascular permeability, increased lung weight, and loss of lung tissue aeration. Typical chest computed tomography findings show images of bilateral ground-glass lung parenchyma with consolidated lung opacities consistent with pulmonary edema. This diffuse alveolar damage, which is the hallmark of the morphology of ARDS, was identified at autopsy in patients with COVID-19 [9,10,11].

According to evidence in the literature, the intensity and urgency of the situation caused by the pandemic led to calls for, and subsequent proliferation of, protocols, tools, and guidance for allocating scarce ICU resources during the COVID-19 pandemic [12], since the length of stay in the ICU has a negative impact on the muscular system, with loss of muscle mass (sarcopenia) and functionality, explained by the association with the patient’s mechanical ventilation (MV) time, on average of 7 days, linked to sepsis and multiple organ failure. Thus, the etiology of the development of muscle weakness is directed towards prolonged immobility, prolonged use of mechanical ventilation, and sepsis, in addition to the use of pharmacological agents such as corticosteroids, muscle relaxants, neuromuscular blockers, glycemic alterations, and the presence of malnutrition [11,13,14,15]. Many patients with COVID-19 who receive treatment in the ICU may demonstrate subsequent severe muscle weakness associated with high functional dependence and neurocognitive disorders that make it difficult to perform activities of daily living (ADL) and may persist for up to five years after hospital discharge [16]. In addition, a large number of COVID-19-specifc protocols and decision aids were released by national, local, and professional bodies to assist with this decision-making. These tools are based on varying combinations of medical, ethical, and social criteria, with a general focus on maximizing the number of lives saved.

COVID-19 in its severe form has also been related to neurological complications, such as critical illness myopathy in 48% to 96% of ICU patients with ARDS, although however, it is usually associated with the use of corticosteroids, immobility, and sepsis, in the presence of myopathy and polyneuropathy, patients affected by COVID-19 present muscle weakness, loss of functionality, and impacts on quality of life that can persist for around two years after the presentation of the condition [16]. Thus, the physiotherapist plays a fundamental role in the rehabilitation process. In addition, these professionals must have experience with critically ill patients in a hospital environment associated with daily planning to guarantee quality and safety for the patient and the team [16].

We are increasingly seeing the development of protocols aimed at avoiding the deleterious effects of immobility, such as early mobilization protocols, especially in ICU environments, as these interventions benefit critically ill patients and reduce the development of acquired muscle weakness [15]. The benefits of early mobilization can be seen throughout the treatment, such as reducing the deleterious effects of the disease, especially on muscle and cardiopulmonary function, and improving the patient’s mobility and functionality [17]. As COVID-19 is a new disease, the impact on long-term outcomes in survivors is still being investigated through scientific studies, particularly concerning the syndrome being denominated “long COVID”, which is characterized by persistent symptoms or long-term symptoms and complications. Hodgson et al. identified, among 115 patients, 6 months after surviving COVID-19, that 71.3% (82 patients) reported persistent symptoms, with shortness of breath, loss of muscle strength, and fatigue being the most frequent symptoms [18].

According to several authors, impaired physical and functional status and symptoms such as dyspnea, desaturation, cough, weakness, and fatigue may persist for weeks after hospital discharge. In addition to the harm caused by hospitalization and/or prolonged inactivity, the persistent high inflammatory burden and previous health conditions seem to negatively influence the recovery of these patients [18,19,20,21,22,23,24,25,26]. Considering that hospitalized patients affected by COVID-19 frequently require a robust rehabilitation program and present severe respiratory and motor impairment, especially those who require intensive treatment and IMV and that the sequelae may extend after the period of hospitalization, the current project aims to investigate the clinical and epidemiological characteristics of patients who required hospitalization in the ICU in the national territory and to measure the severity outcome of these patients, thus collaborating in the implementation of public policies that will help patients affected by COVID-19. Therefore, the aim of this study was to verify the clinical and epidemiological characteristics of patients with COVID-19 admitted to the Intensive Care Unit between 2020 and 2021, according to age group.

## 2. Materials and Methods

Retrospective cohort study. Data were collected in two consecutive years, between March 2020 and April 2021, according to assistance in Intensive Care Units—ICU.

The search for data was carried out through the “ICUs Brasileiras” Project developed by the Associação de Medicina Intensiva Brasileira (AMIB) at the virtual address: http://www.utisbrasileiras.com.br. This project, entitled Brazilian ICUs, is based on the National Registry of Intensive Care, with the objective of characterizing the epidemiological profile of Brazilian Intensive Care Units (ICUs) and sharing epidemiological information that may be useful to guide public health policies in order to develop assistance and research strategies to improve the care of critically ill patients in Brazil. Currently, the project involves 469 hospitals, 1154 adult ICUs, and 22,832 ICU beds (31.6% of the total ICU beds in Brazil). Thus, all patients admitted to the ICUs were evaluated, considering the inclusion and exclusion criteria of the study.

The eligibility criteria for this study were: Patients diagnosed with COVID-19 admitted to the ICUs participating in the Brazilian ICUs Project between March 2020 and November 2021; age groups from 11 years old were included. The following were excluded: patients aged 1–10 years, pregnant women, and patients with chronic neurological or rheumatic diseases prior to the diagnosis of COVID-19.

### 2.1. Procedures and Data Collection

Data on the number of hospitalizations for severe acute respiratory syndrome (SARS) due to COVID-19 were collected between 2020 and 2021. The age groups were analyzed and divided into groups for each decade: from 11 years to 20 years, from 21 years to 30 years, from 31 to 40 years, from 41 to 50 years, from 51 to 60 years, from 61 to 70 years, from 71 to 80 years, from 81 to 90 years, and over 90 years of age.

During the period of the study protocol, the following variables were analyzed: the number of hospital admissions for severe acute respiratory syndrome (SARS) due to COVID-19; demographic information (sex, age group, and presence of comorbidities); length of stay in the ICU; length of stay in the ICU longer than 7 days and longer than 21 days; mean length of hospital stay; and hospital stay longer than 30 days; the use of invasive mechanical ventilation support and non-invasive ventilation; in addition to the mean time of invasive mechanical ventilation [27,28,29,30].

Prognostic scores were also collected, such as SAPS 3 (Simplified Acute Physiologic Score III) and SOFA (Sequential Organ Failure Assessment) and the use of amines and renal support. In addition, we collected information on ICU and hospital mortality rates, non-ventilated patients, ventilated patients, and patients who underwent renal replacement therapy.

### 2.2. Statistical Analysis

All statistical analyses were performed using SPSS version 24 (IBM, Chicago, IL, USA). The Mann–Whitney test was applied for comparisons between 2020 and 2021 for the intervals of months considered in relation to age group, mortality, and the percentage of patients undergoing dialysis and mechanical ventilation. In addition, Kruskal–Wallis analysis of variance was applied to compare age groups in terms of mortality, dialysis patients, and ventilated patients. For all tests, a significance level of 5% was considered [31].

## 3. Results

A total of 166,966 ICU hospital admissions were evaluated over the included period. The main results showed the influence of age group on clinical variables of patients with COVID-19 admitted to the Intensive Care Unit. Table 1 shows the peak number of hospitalizations between March and May 2021 and the variability of percentages between sexes throughout the pandemic period, with a higher percentage of males.

Table 2 presents the percentages of hospital admissions between the different age groups, with no significant differences observed between 2020 and 2021.

Table 3 shows the highest peak percentage of hospital admissions between March and May 2021, for the 7-day interval, and between March and May 2020 presented the highest number of hospital admissions for longer than 21 days for patients diagnosed with COVID-19.

According to Table 4, the mortality percentages between the different age groups did not show significant differences between 2021 and 2021.

Table 5 presents the mortality percentages between the different age groups, with a significant difference shown for the age group above 81 years (*p* < 0.0001), with a higher percentage of mortality than the other age groups. Figure 1 presents the proportions of deaths in the age groups between 40 and 80 years for the months corresponding to 2020 and 2021.

Table 6 presents the mortality percentages between the different age groups of patients with COVID-19 not ventilated in the ICU. No significant differences were observed between 2020 and 2021. Table 7 presents the mortality percentages between the different age groups of patients with COVID-19 who were ventilated in the ICU, which also did not show significant differences between 2020 and 2021.

Table 8 presents the mortality percentages among the different age groups of patients who were ventilated, with a significant difference for the age group above 71 years (*p* < 0.0001), showing a higher percentage of mortality than the other age groups. Table 9 presents the mortality percentages among the different age groups of patients with COVID-19 who underwent dialysis in the ICU. No significant differences were observed between 2021 and 2021.

Table 10 presents the mortality percentages among the patients who underwent dialysis, with a significant difference for the age group above 71 years (*p* < 0.0001) in relation to the other age groups. Figure 2 presents the mean hospital morbidities by SAPS 3 and SOFA of patients with COVID-19 who were admitted to the ICU, showing the predominance of SAPS 3 between 2020 and 2021. Figure 3 presents the percentages of non-invasive ventilation support, mechanical ventilation, amines, and renal support of patients with COVID-19 who were admitted to the ICU, showing the peak of mechanical ventilation with more than 7 days between June and August 2021 and mechanical ventilation together with amines between March and May 2021 (Figure 3). 

## 4. Discussion

COVID-19 is a viral disease that has had devastating effects across the world. To date, there have been more than 6.5 million deaths worldwide. Brazil is one of the countries most affected by the COVID-19 pandemic, with more than 3,451,673 confirmed cases and 684,813 deaths in the national territory, ranking second in the number of deaths in the world [32]. The present study is pioneering in comparing the clinical and epidemiological characteristics between the quarters of 2020 and 2021 of the COVID-19 pandemic.

In the evaluated quarters, a peak in the number of hospitalizations can be observed in the quarter from March to May 2021, which is related to the beginning of the second wave (after epidemiological week 43) of COVID-19, which presented different characteristics from the first wave, with outbreaks in different regions of the country, in addition to great pressure on the health system already impacted by the first wave [33]. During the second wave, in epidemiological week 53, the E484K mutation (Beta variant) was detected, with the first case identified in South Africa, among the SARS-CoV-2 variants in Brazil, present in more than 50% of viral genomes [33].

There was a higher percentage of hospital admissions for males, with a mean of 58.9%, as observed in previously published studies, in which the male sex was prevalent, independently of a higher risk of hospitalization and ICU admission, with a higher risk of death compared to females. This phenomenon can be explained by immunological differences, as demonstrated in a previous study in which men had a greater inflammatory response, a lower percentage of lymphocytes, and a worse antibody response during infection and recovery from SARS-CoV-2. In addition, hormonal and cell composition differences between the sexes may also be related to higher mortality for males [34,35,36].

Another important point was that there were no differences in mortality percentages between the quarters of 2020 and 2021 by age group. However, higher mortality was observed with increasing age groups, corroborating data from the literature, which demonstrate that age is an epidemiological predictor for the outcome of death in this population, with the risk of death doubling every 5 years from childhood onwards [37].

Regarding hospital deaths of patients admitted to the ICUs over the epidemiological weeks, there was a significant reduction in older patients and mortality after the vaccination period, which can be explained by the implementation of the national vaccination program for older adults, which began in January 2021, and which was subsequently offered to all over 30s in June 2021. However, it is also possible to observe an increase in the number of cases in the final quarter in older adults in the age groups sequentially prioritized in the national vaccination program, which sparked research into immunosenescence and the subsequent initiation of booster doses for these populations [38].

There was a gradual increase in younger patients admitted to ICUs in early 2021, which can be explained by the effectiveness of the national vaccination program for older adults that began in January 2021. However, when comparing the percentages of hospital admissions in the three-month intervals between 2020 and 2021 for each age group of patients with COVID-19 admitted to the ICU, no differences were observed between the quarters. In addition, there were higher percentages of hospitalizations in the age group from 61 to 70 years over the first three quarters, which moved to a younger age group in the quarter from March to May 2021, 51 to 60 years, that extended until the June to August quarter of 2021, a fact that has been related to the discovery of the E484K mutation [39].

The highest peak of hospital stays between 7 and 21 days between March and May 2021, related to the second wave of the pandemic, and greater severity in these patients was observed, as seen through the greater use of mechanical ventilation and amines in this quarter. In the comparison between the quarters of 2020 and 2021, regarding the percentage of mortality of ventilated patients, no statistical differences were observed. However, a mortality percentage of ventilated patients above 50% was observed from the age group of 51 to 60 years to the age group above 90 years, data similar to that found by Lim et al. [40] in a meta-analysis that evaluated 69 studies and 57,420 adult patients with COVID-19 who received MV. In addition, the authors found mortality rates from 47.9% in younger patients (age < 40 years) to 84.4% in older patients (age > 80 years), similar to that observed in the current study, with a mean mortality of 85.7% in the age group above 81 years [40].

The mortality percentages of non-ventilated patients between age groups did not vary over the evaluated quarters. However, in the data analyzed, a mean mortality rate of 47.8% was observed in the age group over 90 years in patients not ventilated and admitted to the ICU, with a higher mean percentage of 88.8% in patients aged over 90 years and submitted to MV, corroborating with a systematic review and meta-analysis carried out by Chang et al., [41], which observed that the use of MV, the presence of acute kidney injury, and ARDS correlated with mortality in patients with COVID-19.

When analyzing the mortality percentages between the different age groups of patients with COVID-19 who were dialyzed in the ICU, we found no statistical differences between 2020 and 2021. However, Lin et al. [42] showed in a meta-analysis that patients aged over 60 years and with severe COVID-19 present independent risk factors for acute kidney injury, which increases the risk of death among these patients [42]. In the present study, 50% of patients in the ICU used MV, and 20% used NIV (non-invasive ventilation), over the first three quarters, with a gradual increase in the use of NIV, reflecting the evolution of care for critically ill patients in the ICU [43].

The increase in NIV use was accompanied by a decrease in the percentage of MV use. However, between March and May 2021, there was an increase in MV and NIV use percentages in patients admitted to the ICU, reflecting the severity of the second wave of COVID-19, with 50% on MV for more than 7 days. Greater use of renal replacement therapy was also observed in the first trimester, with a decrease in its use over the following quarters, in addition to a peak in the use of amines in the March to May 2021 quarter. In Brazil, among the 166,961 ICU admissions analyzed, we were able to observe differences in the patterns of outcomes, such as mortality in non-ventilated and ventilated patients, overall ICU mortality, and hospital mortality. Understanding national trends facilitates better resource management and preparation of ICUs, in addition to offering subsidies for better preparation of human resources and health teams needed to confront public health emergencies.

In summary, the main results showed the following points: (a) a peak in the number of hospitalizations in the quarter from March to May 2021, after epidemiological week (EW) 43 of 2020; (b) a higher percentage of hospitalizations in males (mean of 58.9%); (c) we did not observe a difference in the percentage of mortality between the quarters of 2020 and 2021 by age group, but there was a higher percentage of mortality with the increase in the age group over 2020 and 2021; (d) hospital deaths over the epidemiological weeks showed a decrease in the age group over 80 years in EW 8/21 and an increase again in EW 28/21; (e) there was a higher percentage of hospitalizations in the age group 61 to 70 years old in 2020 and 51 to 60 years old in 2021; (f) an increase in the peak of hospital admissions between 7 and 21 days between March and May 2021; (g) there was a higher percentage of mortality (above 50%) among ventilated patients aged between 51 and 90 years; (h) There was a higher percentage of mortality in the age group over 90 years (mean of 47.8%) of non-ventilated patients admitted to the ICU. In those ventilated over 90 years of age, the average was 88.8%; (i) 50% of ICU patients used mechanical ventilation in the first trimester, and around 20% of patients used non-invasive ventilation.

These findings aid in the patient triage process and provide guidance to clinicians for decision-making when allocating critical care resources during the COVID-19 pandemic. According to a study carried out by Aquino et al. [12] in 2022, the main guidelines are (1) the development process; (2) the presence and nature of ethical, medical, and social criteria for allocating critical care resources; and (3) the membership of and decision-making procedure of any triage committees, all based on medical and social criteria including medical need, co-morbidities, prognosis, age, disability and other factors, with a focus on seemingly objective medical criteria. Our analysis reveals that the criteria directed toward the medical (resources) and clinical needs, co-morbidities, prognosis, age, and disability of the patients are highlighted as important points that must be evaluated to assist in the screening process of patients allocated to the ICU, for broad treatment with fundamental clinical resources that preserve the lives of patients [12].

According to Marinelli et al. [43], the most effective and ethically sound response of any national health care system would be to adequately reconfigure its organizational mechanisms and perform clinical trials and all related administrative procedures consistent with the emergency assistance of the COVID-19 patient. In Brazil, the government defined law number 4759/20, which states that during the COVID-19 pandemic, the person diagnosed with the disease has the right to be accompanied by a person of their choice during hospitalization in the ICU. This project is an important guideline, since in the current study, with data from Brazilian ICUs, we showed the high prevalence of males, long ICU stay due to comorbidities and older age, with a high risk of mortality, especially in the year 2021, so having a person accompany the patient in the ICU meets an important ethical precept of respect for the dignity and fragility of the patient.

The limitation of this study was that it did not include data on clinical characteristics, such as liver disease and glomerular filtration rate, variables that could further complement the improvements in care for patients with COVID-19. However, the data from this study are extremely important because they report observations using data from the “ICUs Brasileiras” Project developed by the Associação de Medicina Intensiva Brasileira (AMIB), based on the National Registry of Intensive Care that involves 469 hospitals, 1154 adult ICUs and 22,832 ICU beds (31.6% of the total ICU beds in Brazil), with the objective of characterizing the epidemiological profile of Brazilian intensive care units (ICUs) and sharing epidemiological information that may be useful to guide public health policies and develop assistance and research strategies to improve the care of critically ill patients in Brazil.

## 5. Conclusions

The clinical and epidemiological characteristics of patients with COVID-19 were influenced by age group, showing higher mortality over 81 years and over 70 years in the ICU supported by mechanical ventilation, with hospital stay over 7 days from June to August 2021. The years 2020 and 2021 also showed differences for patients with COVID-19, with greater hospitalizations between March and May 2021, especially in the ICU for 7 days and between March and May 2020 in the 21-day interval.

Hospitalized patients affected by COVID-19 need a robust rehabilitation program, especially those requiring intensive treatment and advanced life support, and the sequelae can extend beyond hospitalization. Thus, these findings can guide doctors and physiotherapists to develop guidelines for patients allocated to ICUs during the COVID-19 pandemic and may contribute to the encouragement and importance of implementing public policies that will support the rehabilitation of patients affected by COVID-19 [19], especially after hospital discharge in the older age group, given the resulting clinical-functional limitations of the disease.

## Figures and Tables

**Figure 1 life-13-00741-f001:**
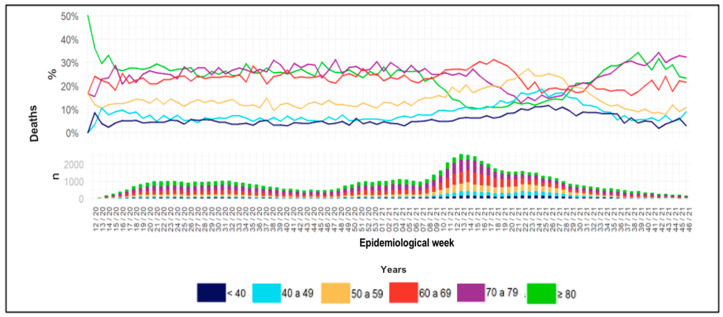
Proportion of hospital deaths of patients diagnosed with COVID-19 according to age group between 2020 and 2021.

**Figure 2 life-13-00741-f002:**
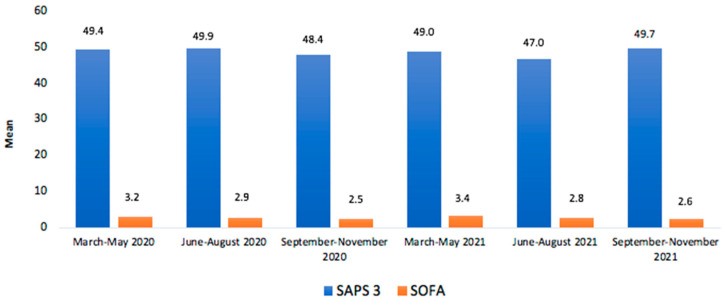
Mean number of hospital morbidities by SAPS 3 and SOFA of patients diagnosed with COVID-19 between the pandemic years 2020 and 2021.

**Figure 3 life-13-00741-f003:**
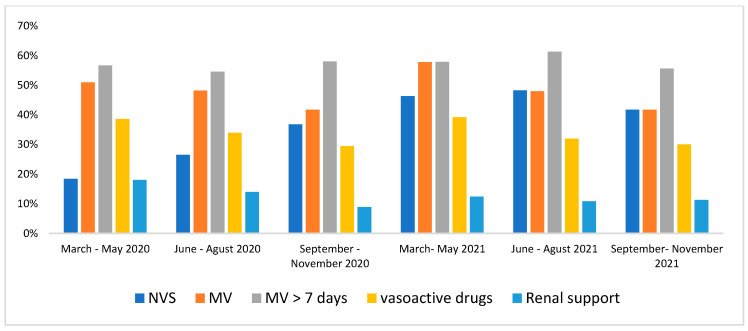
Percentages of non-invasive ventilation support (NVS), mechanical ventilation (MV), vasoactive drugs, and renal support of patients with COVID-19 who were admitted to the ICU of patients with a diagnosis of COVID-19 between the pandemic years 2020 and 2021.

**Table 1 life-13-00741-t001:** Number of hospital admissions and percentage comparisons between sexes, in the three-month intervals between 2020 and 2021 of patients with COVID-19 who were admitted to the Intensive Care Unit.

Variables	Mar.–May 2020	Jun.–Aug. 2020	Sept.–Nov. 2020	Mar.–May 2021	Jun.–Aug. 2021	Sept.–Nov. 2021
Hospitalizations (n)	18,423	29,042	20,646	60,322	31,178	7350
Hospital discharges (n)	18,228	28,423	20,279	58,529	29,869	6016
Male (%)	60.1	58.9	59.6	59.1	60.8	55.4
Female (%)	39.9	41.1	40.4	40.9	39.2	44.6

**Table 2 life-13-00741-t002:** Percentages of hospital admissions between the three-month intervals between 2020 and 2021 for each age group of patients with COVID-19 admitted to the Intensive Care Unit.

Age (Years)	Mar.–May 2020 (%)	Mar.–May 2021 (%)	Jun.–Aug. 2020 (%)	Jun.–Aug. 2021 (%)	Sept.–Nov. 2020 (%)	Sept.–Nov. 2021 (%)
11–20	0.40	0.30	0.50	0.80	0.50	0.70
21–30	3.30	3.20	3.30	5.10	3.20	3.50
31–40	11.00	11.50	9.30	15.30	9.20	7.30
41–50	15.20	18.40	13.90	20.70	13.40	10.80
51–60	19.30	23.30	18.50	21.00	18.50	12.50
61–70	19.90	22.10	22.20	14.00	22.80	21.30
71–80	16.70	14.00	18.50	12.80	18.90	24.90
81–90	11.10	5.90	11.00	8.00	10.90	15.20
>90	3.10	1.30	2.60	2.30	2.40	3.90
Min	0.40	0.30	0.50	0.80	0.50	0.70
Max	19.9	23.3	22.2	21.0	22.8	24.9
Mean	11.1	11.2	11.0	11.1	11.9	11.5
*p* *	0.980		0.980		0.931	

* Mann–Whitney test, considering statistical differences *p* < 0.05.

**Table 3 life-13-00741-t003:** Percentages of hospital admissions for 7- and 21-day periods of patients diagnosed with COVID-19 between the pandemic years 2020 and 2021.

ICU Stay (Days)	Mar.–May 2020 (%)	Jun.–Aug. 2020 (%)	Sept.–Nov. 2020 (%)	Mar.–May 2021 (%)	Jun.–Aug. 2021 (%)	Sept.–Nov. 2021 (%)
>7 days	50.9	49.5	50.1	57.4	54	52.7
>21 days	17.1	15.1	15.6	16.6	16.5	15.6

**Table 4 life-13-00741-t004:** Percentage comparisons of mortality between the three-month intervals in 2020 and 2021 of patients with COVID-19 who were admitted to the Intensive Care Unit.

Age (Years)	Mar.–May 2020 (%)	Mar.–May 2021 (%)	Jun.–Aug. 2020 (%)	Jun.–Aug. 2021 (%)	Sept.–Nov. 2020 (%)	Sept.–Nov. 2021 (%)
11–20	24.7	18.3	19.1	12.4	12.7	17.1
21–30	12.7	18.3	11.9	12.4	11.1	17.1
31–40	11	23	12.3	18.6	10.2	13
41–50	16.3	30.1	17.2	23.8	14	19
51–60	26.4	39.4	27.3	33.7	22.6	27
61–70	40	52.6	41.5	44.3	35.4	35.5
71–80	50	62.2	52.2	50.8	48.1	46
81–90	58	66	59.5	58.2	54.1	53
>90	61.2	64.3	61.9	58.1	57.7	58.7
Min	11.0	18.3	12.0	12.4	11.0	18.3
Max	61.2	66.0	61.9	58.2	61.2	66.0
Mean	33.3	41.5	33.6	34.7	33.3	41.5
*p* *	0.286		0.911		0.530	

* Mann–Whitney test, considering statistical differences *p* < 0.05.

**Table 5 life-13-00741-t005:** Percentage comparisons of mortality by age group by quarter between 2020 and 2021 of patients with COVID-19 who were admitted to the Intensive Care Unit.

Quarters	31–40 Year	41–50 Year	51–60 Year	61–70 Year	71–80 Year	81–90 Year	>90 Year
Mar.–May 2020 (%)	11.0	16.3	26.4	40.0	50.0	58.0	61.2
Jun.–Aug. 2020 (%)	12.3	17.2	27.3	41.5	52.2	59.5	61.9
Sept.–Nov. 2020 (%)	10.2	14.0	22.6	35.4	48.1	54.1	57.7
Dec.–Feb. 2020–2021 (%)	14.8	19.3	28.5	42.1	52.2	59.0	62.0
Mar.–May 2021 (%)	23.0	30.1	39.4	52.6	62.2	66.0	64.3
Jun.–Aug. 2021 (%)	18.6	23.8	33.7	44.3	50.8	58.2	58.1
Sept.–Nov. 2021 (%)	13.0	19.0	27.0	35.5	46.0	53.0	58.7
Mean	14.7	19.9	29.2	41.6	51.6	58.1	60.5
*p*	*p* < 0.001 *						

* Kruskal–Wallis analysis of variance test, considering statistical differences *p* < 0.05.

**Table 6 life-13-00741-t006:** Percentage comparisons of mortality of non-ventilated patients with COVID-19, between the three-month intervals in 2020 and 2021, who were admitted to the Intensive Care Unit.

Age (Years)	Mar.–May 2020 (%)	Mar.–May 2021 (%)	Jun.–Aug. 2020 (%)	Jun.–Aug. 2021 (%)	Sept.–Nov. 2020 (%)	Sept.–Nov. 2021 (%)
11–20	10.6	4.5	2.3	1.4	1.3	3.4
21–30	2.1	3.6	1.7	2.5	1.5	1.2
31–40	1.2	3.2	1.6	2.7	0.9	3.4
41–50	1.4	5.6	2.5	3.5	2.3	4.0
51–60	4.0	7.7	4.5	6.0	3.8	6.9
61–70	7.6	14.1	10.0	10.6	6.6	7.8
71–80	13.7	21.9	16.8	15.6	15.7	15.5
81–90	21.2	34.4	27.8	30.0	24.0	25.8
>90	38.8	45.1	41.8	42.1	37.0	41.8
Min	11.2	31.2	1.6	1.4	0.90	1.2
Max	38.8	45.1	41.8	42.1	37.0	41.8
Mean	11.8	15.7	12.1	12.7	10.3	12.2
*p* *	0.385		0.683		0.474	

* Mann–Whitney test, considering statistical differences *p* < 0.05.

**Table 7 life-13-00741-t007:** Percentage comparisons of mortality of ventilated patients with COVID-19, between the three-month intervals between 2020 and 2021, who were admitted to the Intensive Care Unit.

Age (Years)	Mar.–May 2020 (%)	Mar.–May 2021 (%)	Jun.–Aug. 2020 (%)	Jun.–Aug. 2021 (%)	Sept.–Nov. 2020 (%)	Sept.–Nov. 2021 (%)
11–20	44.1	36.5	42.2	30.7	44.4	50.0
21–30	36.5	47.7	38.3	37.5	44.7	40.7
31–40	31.6	46.5	38.8	42.9	43.3	43.6
41–50	38.8	52.8	44.3	48.9	44.0	52.6
51–60	50.3	61.5	53.5	59.1	53.0	61.8
61–70	61.1	72.6	66.2	69.5	64.8	70.6
71–80	71.8	81.7	76.3	79.4	75.5	81.0
81–90	81.2	87.8	84.8	88.3	84.3	88.2
>90	86.2	92.0	88.0	88.6	87.3	90.8
Min	31.6	36.5	38.3	30.7	43.3	40.7
Max	86.2	90.0	88.0	88.6	87.3	90.8
Mean	55.7	64.3	59.1	60.4	60.1	64.3
*p* *	0.267		0.863		0.730	

* Mann–Whitney test, considering statistical differences *p* < 0.05.

**Table 8 life-13-00741-t008:** Percentage comparisons of mortality of ventilated patients by age group in the quarters of 2020 and 2021 of patients with COVID-19 who were admitted to the Intensive Care Unit.

Quarter	11–20 Year	21–30 Year	31–40 Year	41–50 Year	51–60 Year	61–70 Year	71–80 Year	81–90 Year	>90 Year
Mar.–May 2020 (%)	44.1	36.5	31.6	38.8	50.3	61.1	71.8	81.2	86.2
Jun.–Aug. 2020 (%)	42.0	38.3	38.8	44.3	53.5	66.2	76.3	84.8	88.0
Sept.–Nov. 2020 (%)	44.4	44.7	43.3	44.0	53.0	64.8	75.5	84.3	87.3
Dec.–Feb. 2020–2021 (%)	48.4	43.4	43.2	47.6	57.3	69.2	79.5	85.8	90.8
Mar.–May 2021 (%)	36.5	47.7	46.5	52.8	61.5	72.6	81.7	87.8	92.0
Jun.–Aug. 2021 (%)	30.7	37.5	42.9	48.9	59.1	69.5	79.4	88.3	88.6
Sept.–Nov. 2021 (%)	50.0	40.7	43.6	52.6	61.8	70.6	81.0	88.2	90.8
Median	44.1	40.7	43.2	47.6	57.3	69.2	79.4	85.8	88.6
Mean	42.3	41.2	41.3	47.0	56.6	67.7	77.8	85.7	89.1
*p*	*p* < 0.001 *								

* Kruskal–Wallis analysis of variance test, considering statistical differences *p* < 0.05.

**Table 9 life-13-00741-t009:** Percentage comparisons of the mortality of patients with COVID-19 undergoing dialysis, between the three-month intervals between 2020 and 2021, who were admitted to the Intensive Care Unit.

Age (Years)	Mar.–May 2020 (%)	Mar.–May 2021 (%)	Jun.–Aug. 2020 (%)	Jun.–Aug. 2021 (%)	Sept.–Nov. 2020 (%)	Sept.–Nov. 2021 (%)
11–20	77.8	45.5	50.0	66.7	0.0	100.0
21–30	41.7	64.2	34.2	57.7	55.6	43.8
31–4	47.5	66.3	49.7	61.0	66.1	59.1
41–50	48.6	67.9	59.1	64.6	56.0	53.4
51–60	60.4	76.0	65.0	72.3	63.1	67.9
61–70	66.9	81.2	74.3	74.9	74.5	68.2
71–80	77.7	86.1	81.0	85.4	85.4	84.7
81–90	85.4	88.7	86.1	89.0	88.1	88.4
>90	86.8	96.8	86.2	91.3	90.0	95.0
Min	41.7	45.5	34.2	57.7	0.0	43.8
Max	86.8	96.8	86.2	91.3	90.0	100.0
Mean	65.8	74.7	59.1	73.6	64.3	73.4
*p* *	0.340		0.321		0.604	

* Mann–Whitney test, considering statistical differences *p* < 0.05.

**Table 10 life-13-00741-t010:** Percentage comparisons of mortality of dialysis patients by age group in the quarters of 2020 and 2021 of patients with COVID-19 who were admitted to the Intensive Care Unit.

Quarters	11–20 Year	21–30 Year	31–40 Year	41–50 Year	51–60 Year	61–70 Year	71–80 Year	81–90 Year	>90 Year
Mar.–May 2020 (%)	77.8	41.7	47.5	48.6	60.4	66.9	77.7	85.4	86.8
Jun.–Aug. 2020 (%)	50.0	34.20	49.7	59.1	65.0	74.3	81.0	86.1	86.1
Sept.–Nov. 2020 (%)	0	55.6	66.1	56.0	63.1	74.5	85.4	88.1	90.0
Dec.–Feb. 2020–2021 (%)	50.0	48.0	57.5	61.1	72.4	78.3	84.6	88.5	94.1
Mar.–May 2021 (%)	45.5	64.2	66.3	67.9	76.0	81.200	86.1	88.7	96.8
Jun.–Aug. 2021 (%)	66.7	57.7	61.0	64.6	72.3	74.900	85.4	89.0	91.3
Sept.–Nov. 2021 (%)	100.0	43.8	59.1	53.4	67.9	68.200	84.7	88.4	95.0
Mean	55.7	49.3	58.1	58.6	68.1	74.0	83.5	87.7	91.4
*p*	*p* < 0.001 *								

* Kruskal–Wallis analysis of variance test, considering statistical differences *p* < 0.05.

## Data Availability

Data can be requested by email to the corresponding author: apribeiro@alumni.usp.br.

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
