# Peer review of "Clinical and Epidemiological Characteristics of Patients with COVID-19 Admitted to the Intensive Care Unit: A Two-Year Retrospective Analysis"

_life, 2023, doi:10.3390/life13030741_

Round 1

Reviewer 1 Report

Abstract: the purpose of the article is clear. I would summarize the section relating to methods, then developed in the specific section. Results and conclusions are clear and specific.

Introduction: the chapter is quite complete and exhaustive, my only comment about the number of citations, I believe that authors need to enrich their manuscript with more citations from updated and recent papers, in particular for these two paragraphs:

1- Approximately 20% of individ- 56 uals with COVID-19 develop the severe form of the disease, requiring hospitalization and 57 oxygen therapy support, associated with lung injury and hypoxemic respiratory failure. 58 Around 5% are admitted to the Intensive Care Unit (ICU), where they need invasive me- 59 chanical ventilation for the treatment of ARDS [6]. Patients admitted to the ICU can be 60 evaluated using several scales, developed to quantify the severity of the disease, assess its 61 prognosis, and direct better therapeutic interventions. Please, add:

-Prevention and Treatment of Life-Threatening COVID-19 May Be Possible with Oxygen Treatment. Life 202212, 754. https://doi.org/10.3390/life12050754

- Different Methods to Improve the Monitoring of Noninvasive Respiratory Support of Patients with Severe Pneumonia/ARDS Due to COVID-19: An Update. J Clin Med. 2022;11(6):1704. Published 2022 Mar 19. doi:10.3390/jcm11061704

2- The consequences caused by COVID-19 are diverse, affecting several systems, with 67 the respiratory system being one of the most commonly affected, regardless of the symp- 68 tomatology. Respiratory sequelae are very varied and depend on the type of injury, asso- 69 ciated infections, and presence of previous pulmonary comorbidities, highlighting a re- 70 duction in lung capacity and volume, changes in radiographic findings, limitations in the 71 performance of exercises, and consequently, a decrease in functional capacity

-Can Lung Imaging Scores and Clinical Variables Predict Severe Course and Fatal Outcome in COVID-19 Pneumonia Patients? A Single-Center Observational Study. Life 202212, 735. https://doi.org/10.3390/life12050735

- Interstitial Lung Disease at High Resolution CT after SARS-CoV-2-Related Acute Respiratory Distress Syndrome According to Pulmonary Segmental Anatomy. J Clin Med. 2021;10(17):3985. 

Materials and methods: the section is complete and exhaustive. Were any patients excluded from the study? If yes, how many and why?

Results: the results are displayed in a clear and complete manner. I appreciated the presence of tables and graphs that facilitate consultation. If possible, I would add colors to the tables to make them easier to read.

Discussion: according to what is stated in the results, obtained data are commented on in a precise and timely manner. Also in this case I would add a summary table or diagram for better consultation.

Conclusions: they briefly summarize what has been obtained. I would add a section related to future perspectives or ideas on how to improve/expand the collected results.

Author Response

São Paulo, 07th of January 2023

Life

Dear Chief Editor,

Prof. Dr. William Bains

Editor and Editorial Board Member of LIFE

We, the authors, would like to resubmit the paper Clinical and epidemiological characteristics of patients with COVID-19 admitted to the Intensive Care Unit: a two-year retrospective analysis” (life-2144981) in a revised form, as suggested. We are sending also a covering letter responding to the reviewer’s comments on a point-by-point basis. Bellow, we followed your comments and answered each one also in a point-by-point basis. The authors would like to thank the editor for the careful revision and constructive comments/suggestions on our manuscript that certainly contributed for a better version of it.

Yours sincerely,

The authors

Responses to the reviewers' comments on a point-by-point basis

Submission: Ref: Submission (life-2144981)

Reviewer #1:

We thank the reviewer for the constructive comments on our manuscript. Your suggestions and remarks have helped us to reflect on the manuscript and make a better version. We appreciate your suggestions and carefully considered every one of your comments and we made the appropriate changes. Below, we responded to your remarks on a point-by-point basis and inserted the corrections to the new version of the manuscript (underlined parts). Some of your comments will be discussed here. First your comment is given in bold; subsequently we provide our answer.

Reviewer's report

English language and style are fine/minor spell check required.

Answer: We appreciated your comment and strongly apologize for the English language in the manuscript. We have submitted the manuscript to a professional service for reviewing English as a second language (Scribendi). We hope it achieves the standards of this respectful Journal (Journal of Clinical Medicine).

Comments and Suggestions for Authors

  1. Abstract: the purpose of the article is clear. I would summarize the section relating to methods, then developed in the specific section. Results and conclusions are clear and specific.

Answer: We thank the reviewer for work and help in the best manuscript. We appreciate your suggestion and we have made the methods section more concise. Page 1 (line 17-22). Underlined parts. 

  1. - Introduction: the chapter is quite complete and exhaustive, my only comment about the number of citations, I believe that authors need to enrich their manuscript with more citations from updated and recent papers, in particular for these two paragraphs:

1- Approximately 20% of individ- 56 uals with COVID-19 develop the severe form of the disease, requiring hospitalization and 57 oxygen therapy support, associated with lung injury and hypoxemic respiratory failure. 58 Around 5% are admitted to the Intensive Care Unit (ICU), where they need invasive me- 59 chanical ventilation for the treatment of ARDS [6]. Patients admitted to the ICU can be 60 evaluated using several scales, developed to quantify the severity of the disease, assess its 61 prognosis, and direct better therapeutic interventions. Please, add:

- Prevention and Treatment of Life-Threatening COVID-19 May Be Possible with Oxygen Treatment. Life 2022, 12, 754. https://doi.org/10.3390/life12050754

- Different Methods to Improve the Monitoring of Noninvasive Respiratory Support of Patients with Severe Pneumonia/ARDS Due to COVID-19: An Update. J Clin Med. 2022;11(6):1704. Published 2022 Mar 19. doi:10.3390/jcm11061704

Answer: We greatly appreciate your attention and work in suggesting this scientific evidence to us. We have added information from this scientific evidence in the text and added suggested references. Page 2 (line 63-69); Page 14 (line 466-472). Underlined parts.

  1. 2-The consequences caused by COVID-19 are diverse, affecting several systems, with 67 the respiratory system being one of the most commonly affected, regardless of the symp- 68 tomatology. Respiratory sequelae are very varied and depend on the type of injury, asso- 69 ciated infections, and presence of previous pulmonary comorbidities, highlighting a re- 70 duction in lung capacity and volume, changes in radiographic findings, limitations in the 71 performance of exercises, and consequently, a decrease in functional capacity

-Can Lung Imaging Scores and Clinical Variables Predict Severe Course and Fatal Outcome in COVID-19 Pneumonia Patients? A Single-Center Observational Study. Life 2022, 12, 735. https://doi.org/10.3390/life12050735

- Interstitial Lung Disease at High Resolution CT after SARS-CoV-2-Related Acute Respiratory Distress Syndrome According to Pulmonary Segmental Anatomy. J Clin Med. 2021;10(17):3985.

Answer: We appreciate your comment and strongly agree. We have added information from this scientific evidence in the text and added suggested references. Page 2 (line 75-81); Page 14 (line 466-472). Underlined parts.

  1. Materials and methods: the section is complete and exhaustive. Were any patients excluded from the study? If yes, how many and why?

Answer: We appreciate your comment and thank all your work to improve the manuscript. However, we would like to reiterate that this search for data was carried out through the “ICUs Brasileiras” Project developed by the Associação de Medicina Intensiva Brasileira (AMIB) at the virtual address: http://www.utisbrasileiras.com.br. This project, entitled Brazilian ICUs, is based on the National Registry of Intensive Care, with the objective of characterizing the epidemiological profile of Brazilian intensive care units (ICUs) and sharing epidemiological information that may be useful to guide public health policies, to develop assistance and research strategies to improve the care of critically ill patients in Brazil. Currently, the project involves 469 hospitals, 1,154 adult ICUs and 22,832 ICU beds (31.6% of the total ICU beds in Brazil). Thus, all patients admitted to the ICUs were evaluated, considering the inclusion and exclusion criteria of the study. The eligibility criteria for this study were: Patients diagnosed with COVID-19 ad-mitted to the ICUs participating in the Brazilian ICUs Project between March 2020 and November 2021. Age groups from 11 years old were included. The following were excluded: patients aged 1-10 years, pregnant women and patients with chronic neurological or rheumatic diseases, prior to the diagnosis of COVID-19.

This information was better clarified in the text. Page 4 (line 163-170). Underlined parts.

  1. Results: the results are displayed in a clear and complete manner. I appreciated the presence of tables and graphs that facilitate consultation. If possible, I would add colors to the tables to make them easier to read.

Answer: We really appreciate the reviewer and agree with his comment. Therefore, we added color to the tables to make them easier to read. Page 5-10 (Tables).

  1. Discussion: according to what is stated in the results, obtained data are commented on in a precise and timely manner. Also in this case I would add a summary table or diagram for better consultation.

Answer: We really appreciate and agree with the reviewer, so we have added a summary of the main points observed in the present study at the end of the discussion for better consultation. We only left it in writing to also answer the question of reviewer 2. In addition, was added the limitation of the study. We hope to have answered your suggestion. Page 12 (line 491-405) and Page 12 (line 406-410). Underlined parts.

  1. Conclusions: they briefly summarize what has been obtained. I would add a section related to future perspectives or ideas on how to improve/expand the collected results.

Answer: I have no words to thank you for your comments, which, without a doubt, helped us to make the information in the manuscript clearer, more coherent and with an improvement in scientific writing based on the literature. We would like to reiterate our thanks for your work, efficiency and attention in helping us. Thus, as suggested by the reviewer, we added the clinical relevance and future perspective of the present study. Page 13 (line 419-425). Underlined parts.

Reviewer 2 Report

I read with great interest the paper “Clinical and epidemiological characteristics of patients with COVID-19 admitted to the Intensive Care Unit: a two-year retrospective analysis” by Juliana Raimondo e Silva Malzone et al.

The design is fine. The article is logically divided into sections and subsections. Data presented are of good interest.

Major Comments:

1.      Introduction is way too long, and several data reported are not needed. Please revise.

2.      The title is misleading. In fact, the authors only reported data about gender and age, not treating other clinical characteristics such as hepatopathy, glomerular filtration rate, other comorbidities. Please revise the manuscript accordingly by adding such data if possible. And discuss it in the appropriate section.

3.      Please add limitation of the study at the end of discussion.

Author Response

São Paulo, 07th of January 2023

Life

Dear Chief Editor,

Prof. Dr. William Bains

Editor and Editorial Board Member of LIFE

We, the authors, would like to resubmit the paper Clinical and epidemiological characteristics of patients with COVID-19 admitted to the Intensive Care Unit: a two-year retrospective analysis” (life-2144981) in a revised form, as suggested. We are sending also a covering letter responding to the reviewer’s comments on a point-by-point basis. Bellow, we followed your comments and answered each one also in a point-by-point basis. The authors would like to thank the editor for the careful revision and constructive comments/suggestions on our manuscript that certainly contributed for a better version of it.

Yours sincerely,

The authors

Responses to the reviewers' comments on a point-by-point basis

Submission: Ref: Submission (life-2144981)

Reviewer #2:

We thank the reviewer for the constructive comments on our manuscript. Your suggestions and remarks have helped us to reflect on the manuscript and make a better version. We appreciate your suggestions and carefully considered every one of your comments and we made the appropriate changes. Below, we responded to your remarks on a point-by-point basis and inserted the corrections to the new version of the manuscript (underlined parts). Some of your comments will be discussed here. First your comment is given in bold; subsequently we provide our answer.

Reviewer's report

Moderate English changes required

Answer: We appreciated your comment and strongly apologize for the English language in the manuscript. We have submitted the manuscript to a professional service for reviewing English as a second language (Scribendi). We hope it achieves the standards of this respectful Journal (Journal of Clinical Medicine).

Comments and Suggestions for Authors

  1. I read with great interest the paper “Clinical and epidemiological characteristics of patients with COVID-19 admitted to the Intensive Care Unit: a two-year retrospective analysis” by Juliana Raimondo e Silva Malzone et al. The design is fine. The article is logically divided into sections and subsections. Data presented are of good interest.

Answer: We greatly appreciate your comments, help, and all the work you do to help us improve the manuscript. Thank you very much for your excellent work.

  1. Major Comments: 1. Introduction is way too long, and several data reported are not needed. Please revise.

Answer: We really appreciate and agree with your excellent comment. Thus, we have revised the introduction and synthesized some paragraphs and removed some information and replaced it with more recent scientific evidence information. In this way, we added the following references below. Page 2 (line 53-54; 63-69; 75-81); Page 3 (line 114-135); and Page 14 (line 466-476). Underlined parts.

- Prevention and Treatment of Life-Threatening COVID-19 May Be Possible with Oxygen Treatment. Life 2022, 12, 754. https://doi.org/10.3390/life12050754.

- Different Methods to Improve the Monitoring of Noninvasive Respiratory Support of Patients with Severe Pneumonia/ARDS Due to COVID-19: An Update. J Clin Med. 2022;11(6):1704. Published 2022 Mar 19. doi:10.3390/jcm11061704.

- Can Lung Imaging Scores and Clinical Variables Predict Severe Course and Fatal Outcome in COVID-19 Pneumonia Patients? A Single-Center Observational Study. Life 2022, 12, 735. https://doi.org/10.3390/life12050735

- Interstitial Lung Disease at High Resolution CT after SARS-CoV-2-Related Acute Respiratory Distress Syndrome According to Pulmonary Segmental Anatomy. J Clin Med. 2021;10(17):3985.

  1. The title is misleading. In fact, the authors only reported data about gender and age, not treating other clinical characteristics such as hepatopathy, glomerular filtration rate, other comorbidities. Please revise the manuscript accordingly by adding such data if possible. And discuss it in the appropriate section.

Answer: We really appreciate with your excellent comment. However, during the period of the study protocol, the following variables were analyzed: the number of hospital admissions for Severe Acute Respiratory Syndrome (SARS) due to COVID-19; demographic information (sex, age group, and presence of comorbidities); the length of stay in the ICU, the length of stay in the ICU longer than 7 days and longer than 21 days; mean length of hospital stay and hospital stay longer than 30 days; the use of invasive mechanical ventilation support and non-invasive ventilation, in addition to the mean time of invasive mechanical ventilation, SAPS 3 (Simplified Acute Physiologic Score III), SOFA (Sequential Organ Failure Assessment), and the use of Amines and Renal Support. In addition, we collected information on: ICU mortality rate and hospital mortality rate, non-ventilated patients, ventilated patients, and patients who underwent renal replacement therapy. The reason for collecting only these variables was access to the database available in “ICUs Brasileiras” Project developed by the Associação de Medicina Intensiva Brasileira (AMIB) at the virtual address: http://www.utisbrasileiras.com.br. So, unfortunately not included in the variables suggested by the reviewer. This information was better described in the methods section. Page 4 (line 176-182). Underlined parts.

  1. Please add limitation of the study at the end of discussion.

Answer: We are very grateful and agree with your suggestion. We have added the study limitation in the discussion section, taking into account your previous query. Page 12 (line 406-410). Underlined parts. 

Round 2

Reviewer 2 Report

Unfortunately, the authors did not manage to improve the manuscript as I expected. English must be revised by a native speaker. The lack of clinical characteristics is an important limitation and in contrast with the title. I suggest the authors to improve the manuscript, provide more data and eventually resubmit. 

Author Response

São Paulo, 04th of February 2023

Life

Dear Chief Editor,

Prof. Dr. William Bains

Editor and Editorial Board Member of LIFE

We, the authors, would like to resubmit the paper Clinical and epidemiological characteristics of patients with COVID-19 admitted to the Intensive Care Unit: a two-year retrospective analysis” (life-2144981) in a revised form, as suggested. We are sending also a covering letter responding to the reviewer’s comments on a point-by-point basis. Bellow, we followed your comments and answered each one also in a point-by-point basis. The authors would like to thank the editor for the careful revision and constructive comments/suggestions on our manuscript that certainly contributed for a better version of it.

Yours sincerely,

The authors

Responses to the reviewers' comments on a point-by-point basis

Submission: Ref: Submission (life-2144981)

Reviewer #2:

We thank the reviewer for the constructive comments on our manuscript. Your suggestions and remarks have helped us to reflect on the manuscript and make a better version. We appreciate your suggestions and carefully considered every one of your comments and we made the appropriate changes. Below, we responded to your remarks on a point-by-point basis and inserted the corrections to the new version of the manuscript (underlined parts). Some of your comments will be discussed here. First your comment is given in bold; subsequently we provide our answer.

Reviewer's report

Extensive editing of English language and style required

Answer: We appreciated your comment and strongly apologize for the English language in the manuscript. We have submitted the manuscript to a professional service for reviewing English as a second language (Scribendi). We hope it achieves the standards of this respectful Journal (Life)

Comments and Suggestions for Authors

  1. Unfortunately, the authors did not manage to improve the manuscript as I expected. English must be revised by a native speaker. The lack of clinical characteristics is an important limitation and in contrast with the title. I suggest the authors to improve the manuscript, provide more data and eventually resubmit.

Answer: We greatly appreciate your comments, help, and all the work you do to help us improve the manuscript. The limitation of the study has been added in the text, as suggested and the clinical characteristics in the ICU were further explored in the discussion for better understanding. We hope we met the reviewer's expectations.

Page 12 (line 414-420); Page 13 (line 421-447). Underlined parts. 
